# Needle-Free Jet Injection of Poly-(Lactic Acid) for Atrophic Acne Scars: Literature Review and Report of Clinical Cases

**DOI:** 10.3390/jcm13020440

**Published:** 2024-01-13

**Authors:** Nark-Kyoung Rho, Hyun-Jo Kim, Hyun-Seok Kim, Won Lee

**Affiliations:** 1Leaders Aesthetic Laser & Cosmetic Surgery Center, Seoul 06014, Republic of Korea; 2CNP Skin Clinic, Seoul 06030, Republic of Korea; 3Kim Hyun Seok Plastic Surgery Clinic, Seoul 06030, Republic of Korea; 4Yonsei E1 Plastic Surgery Clinic, Seoul 06030, Republic of Korea

**Keywords:** acne scars, atrophic scars, drug-delivery systems, needle-free injectors, poly-(lactic acid), poly-(D,L-lactic acid), poly-(L-lactic acid)

## Abstract

Acne scars, particularly atrophic ones, present a persistent challenge in cosmetic medicine and surgery, requiring extended and multifaceted treatment approaches. Poly-(lactic acid) injectable fillers show promise in managing atrophic acne scars by stimulating collagen synthesis. However, the utilization of needle-free injectors for delivering poly-(lactic acid) into scars remains an area requiring further exploration. In this article, a summary of the latest advancements in needle-free jet injectors is provided, specifically highlighting the variations in jet-producing mechanisms. This summary emphasizes the differences in how these mechanisms operate, offering insights into the evolving technology behind needle-free injection systems. The literature review revealed documented cases focusing on treating atrophic acne scars using intralesional poly-(lactic acid) injections. The results of these clinical studies could be supported by separate in vitro and animal studies, elucidating the feasible pathways through which this treatment operates. However, there is limited information on the use of needle-free jet injectors for the intradermal delivery of poly-(lactic acid). Clinical cases of atrophic acne scar treatment are presented to explore this novel treatment concept, the needle-free delivery of poly-(lactic acid) using a jet pressure-based injector. The treatment demonstrated efficacy with minimal adverse effects, suggesting its potential for scar treatment. The clinical efficacy was supported by histological evidence obtained from cadaver skin, demonstrating an even distribution of injected particles in all layers of the dermis. In conclusion, we suggest that novel needle-free injectors offer advantages in precision and reduce patient discomfort, contributing to scar improvement and skin rejuvenation. Further comprehensive studies are warranted to substantiate these findings and ascertain the efficacy of this approach in scar treatment on a larger scale.

## 1. Introduction

Clinical treatments for atrophic acne scars encompass punch excision, subcision, dermabrasion, chemical peeling, fractional lasers and radiofrequency, and injectable fillers [1]. Among the various materials available for injectable fillers, poly-(lactic acid) (PLA) has been examined for the treatment of atrophic acne scars and was found to be a favorable choice for correcting macular atrophic scarring in individuals with thin dermal tissue. PLA injection is particularly noteworthy compared to other commonly used injectable hyaluronic acid (HA) fillers for treating atrophic acne scars [2]. However, there is a scarcity of reports detailing the treatment of atrophic acne scars through the intradermal delivery of PLA without relying on conventional sharp needles. This article provides an overview of the mechanism by which PLA improves depressed scars, particularly atrophic acne scars. It also examines the recent advancements in needle-free jet injectors and their application in delivering PLA to acne scar lesions. To affirm its feasibility, the article presents clinical cases of atrophic acne scars, demonstrating the intradermal delivery of PLA using an innovative needle-free injector.

## 2. Poly-(Lactic Acid) Injection for Atrophic Acne Scars

### 2.1. Fillers to Treat Acne Scars

Atrophic scars are characterized by volumetric deficiencies and typically require longer treatment periods [3]. Numerous approaches have been explored for managing atrophic acne scarring. Laser treatments (both fractional and non-fractional, encompassing ablative and non-ablative techniques), chemical peels (comprising glycolic acid, trichloroacetic acid, and Jessner’s solution), microneedling, platelet-rich plasma as an adjunctive therapy, and dermal filler injections have been extensively investigated, backed by strong evidence-based support [4]. Among them, filler injections have gained widespread clinical acceptance and demonstrated promising outcomes for atrophic scar treatment [5]. Various filler options, including collagen-based products, fat transfer, and silicon, are available, with hyaluronic acid (HA) being the most frequently used material for localized injections [4]. Fillers prove most effective for soft, rolling acne scars and can be directly injected beneath individual scars or applied broadly to regions with skin laxity or deep tissue atrophy [6]. Combining fillers with other treatments can augment outcomes [4]. For instance, in a randomized control trial encompassing three groups of 20 patients each, the cohort receiving an HA filler with subcision displayed substantial improvements according to qualitative scar scale scores compared to subcision alone, exhibiting results akin to those achieved using a fractional laser plus subcision [7]. Transient side effects of dermal fillers include pain, edema, and erythema at the injection sites [8], but severe complications like vascular compromise leading to subsequent skin necrosis and potential visual loss can rarely occur.

### 2.2. Injectable Poly-(Lactic Acid) for Atrophic Acne Scars

Polylactide, often abbreviated as PLA, represents a thermoplastic aliphatic polyester that showcases varied properties contingent upon its stereochemical configurations [9]. Two primary types of PLA commonly used in medical and surgical applications are poly-(L-lactic acid) (PLLA) and poly-(D,L-lactic acid) (PDLLA), a copolymer derived from the L- and D-forms of PLA. The stereochemistry of PLA isomers notably influences their crystallinity and material characteristics: PLLA exhibits semicrystalline properties, whereas PDLLA primarily remains amorphous [9]. Due to its biocompatibility, biodegradability, and minimal immunological reactivity [10], PLA has emerged as a non-surgical substitute for facial fat grafting. When injected into the subcutaneous layer, PLA generates gradual volume and promotes facial rejuvenation over an extended period [11]. PLA also corrects the contours of depressed scars by stimulating neocollagenesis [6], while the primary function of most dermal fillers in addressing atrophic scars is to enhance volume [4].

In an initial open-label study, 20 individuals with facial scarring attributed to acne or varicella underwent a series of injectable PLLA treatments administered at monthly intervals across seven sessions. Following injections, a notable reduction in scar size was observed, and patient satisfaction with the treatment displayed an increasing trend with each session [12]. In a Canadian study involving 22 patients, PLLA was evaluated for its effectiveness and safety in treating “hill and valley” acne scarring related to aging skin. Over three to four treatments, PLLA injections notably improved scarring for up to 68.2% of participants, enhancing treatment satisfaction by 44%, and exhibited a good safety profile [13]. Serial injections of PLLA have demonstrated significant improvement in challenging atrophic acne scars among patients who previously underwent unsuccessful treatments, such as CO_2_ laser resurfacing, dermabrasion, chemical peels, and various dermal fillers [14]. Patients tend to show increasing satisfaction with serial PLA injections [14], notably improving from satisfactory to better ratings by the follow-up visit, with the most substantial increase observed between Sessions 2 and 4 [13,14]. Recent approaches in treating acne scars involve combining microneedle fractional radiofrequency with the intradermal delivery of PLA [1,15]. Most available reports and studies on PLA for acne scars remain preliminary, aiming to establish the efficacy and safety of this approach. However, to validate or refute these preliminary findings, well-designed controlled studies with higher objectivity and larger patient cohorts are essential [16].

### 2.3. Mechanism of Action

The growing utilization of PLA in treating acne scars relies on its well-documented biological mechanism for stimulating collagen synthesis, supported in the literature [10]. Figure 1 outlines the proposed pathophysiology of atrophic acne scarring along with the role of PLA. After PLA injection, collagen synthesis initiates within 6 to 8 weeks, with the ongoing production of type 1 collagen persisting for 9 to 12 months following the treatment [17]. PLA boosts fibroblast proliferation and concurrently suppresses collagen-degrading enzymes, culminating in increased collagen and elastin content in aging skin [18]. The histological analysis of in vivo human skin subsequent to PDLLA injection revealed notably elevated levels of collagen and elastic fibers within the dermis [19]. An initial rise in Col1α1, Col3α1, TGF-β1, TGF-β2, and TGF-β3 was observed two weeks after injecting PLLA, followed by a decreases observed at the twelve-week timepoint [20]. Macrophage reactions to PLA affect subsequent fibroblast actions and collagen synthesis. In vitro, PLLA triggers an inflammatory response, increasing various inflammation-related cytokines like CCL1, TNFRII, MIP-1α, and IL-8 in M1 macrophages, and MIP-1α and MIP-1β in M2 macrophages, in comparison to calcium hydroxylapatite and unstimulated controls [21]. Injections of PLLA have demonstrated the ability to induce M2 macrophage polarization and elevate the expression of essential factors such as IL-4, IL-13, and TGF-β for collagen synthesis [18]. PDLLA injection enhanced collagen synthesis in an animal model by influencing macrophages to increase NRF2 expression, leading to adipose-derived stem cell proliferation and the secretion of TGF-β and FGF2, which eventually resulted in increased collagen synthesis and potentially mitigated age-related soft tissue volume loss [22]. To summarize, PLA functions by eliciting a regulated inflammatory response, thereby promoting neocollagenesis [23]. Recent studies have additionally shown that apart from stimulating collagen synthesis, PLA also functions as a promoter of angiogenesis [24] and an immunomodulator [24] during the wound healing process. Injectable PLLA and PDLLA, despite sharing biostimulatory properties, differ in collagen formation mechanisms and particle morphology, leading to distinct early-stage volume effects. PLLA exhibits increasing volume effects over time, while PDLLA maintains consistent effects due to differing patterns of neo-tissue growth [25].

## 3. Needle-Free Delivery of Poly-(Lactic Acid) Using Jet Injectors

### 3.1. Intradermal Injections Using Hypodermic Needles

In conventional intradermal injection procedures, manual administration using syringes and needles is typical. This process involves a multi-puncture technique to deliver small amounts of filler into the superficial dermal layer [26]. Although this method is straightforward and cost-effective, it presents several drawbacks, including procedural discomfort, patient apprehension related to needles, the possibility of inconsistent and imprecise injections, and a prolonged treatment duration [27]. Approaches utilizing multi-needle injectors have been introduced to address and alleviate these limitations, providing improved accuracy and stability compared to manual injections. However, they are still burdened by the pain and discomfort caused by needle penetration into the skin, which patient needle-phobia can exacerbate [28].

### 3.2. Laser-Assisted Needle-Free Jet Injectors

To address the limitations of needle-based injections, “no-needle injection” devices were developed, exemplified by the latest needle-free jet injectors powered by laser energy [29]. This category of injector utilizes laser pulses to produce vapor bubbles, generating pressure energy for propelling a jet through a nozzle. A notable advantage of the laser-induced microjet injector lies in its precise energy deposition control, enabling the accurate injection of µL-range volumes to specific depths across the superficial to deep dermal layers [30]. In a recent study, Han et al. [31] conducted a randomized, double-blinded trial, which employed a split-face design, to compare the outcomes of using the laser-induced microjet injector (MIRAJET; JSK Biomed, Seoul, Republic of Korea) versus conventional needle injection for delivering PLA filler to achieve skin enhancement and rejuvenation. A single treatment session yielded comparable enhancements in skin moisture content and elasticity, as seen in traditional manual needle injection, but with fewer adverse effects, a reduced treatment duration, and shorter recovery periods. Laser-driven needle-free injection offers precise filler administration for skin rejuvenation, resulting in even distribution, improved clinical effectiveness, reduced discomfort, and fewer side effects, making it a promising alternative for filler injection [31]. A drawback of laser-based jet injectors is their lower cost-effectiveness, as they require laser systems with substantially higher costs for operation and maintenance [30].

### 3.3. Electromechanical Needle-Free Jet Injectors

Laser-assisted needle-free injection also has disadvantages in scar treatment since it typically has restricted penetration depth in thick and fibrous tissue [30]. For instance, after being injected using a laser-assisted jet injector, PLA was discovered to be localized only in the uppermost part of the dermis (papillary dermis), sparing the mid- and lower dermis [27]. Given the necessity to inject dermal fillers into dense fibrotic areas to relax collagen fiber arrangement for improving atrophic acne scars [32], the principal drawback of laser-driven jet injectors lies in their restricted penetration depth, posing a challenge for scar treatment in these cases. Researchers have explored alternative power sources to enhance the penetration depth of jet injectors and discovered that electromagnetic forces can effectively achieve this goal. This method accelerates liquid movement by utilizing an electromechanical actuator to regulate the piston, a method also referred to as Lorentz force-driven jet injectors [33]. Electromagnetic force enables the deeper penetration of a thick and viscous liquid jet, often reaching several millimeters, while pulsed lasers are more suitable for shallower depths, typically less than 1 mm [30,34]. To summarize, both laser-driven and electromagnetic-based jet injectors possess advantages and disadvantages in comparison to each other. Laser-driven jet injectors may be preferred for precise injection control, particularly in procedures such as skin rejuvenation targeting facial wrinkles and pores. In terms of injection depth and delivery volume, the electromagnetic system may be deemed more suitable, especially for procedures like treating the scalp or scar tissue.

### 3.4. Advantages of Needle-Free Jet Injectors for Acne Scar Treatment

One of the most significant advantages of needle-free pressure injectors compared to manual injections could be their comparatively lower pain levels. Comparative studies directly assessing manual injection versus jet-injection techniques are available in the literature. In a recent comparative study, 17 patients received injections with a needle, while 22 patients underwent jet injector injections. The study showed significant differences in patient-reported pain levels during the procedure (*p* < 0.005), with those in the jet injector group experiencing less pain than those receiving needle injections [35]. A split-face comparison study demonstrated that both manual injections and jet-injections of PLA resulted in similar skin rejuvenation effects, while the use of jet injectors showed superiority in terms of reduced pain, downtime, and side effects [31]. Another advantage of jet injectors is their ability to precisely control the amount of injection. Given that the formation of nodules or granulomas is a frequently reported side effect associated with the intradermal injection of dermal filler [36], especially after injecting PLA [37], the precise control of the injection dose is of paramount clinical importance. It is widely acknowledged that precision in PLA injections into facial skin is crucial as even minor errors like uneven material distribution can lead to irregular outcomes [38]. Researchers suggest using the lowest volumes as possible (e.g., injections of less than 0.05 mL) of more diluted injectable PLA to avoid such problems [38,39]. Given that novel needle-free jet injectors are engineered to deliver precise drug amounts (e.g., 0.0015 mL per shot) to a specific depth, the anticipated risk of implant nodule formation with this system might be considerably lower compared to manual syringe injections.

## 4. Presentation of Clinical Cases

### 4.1. Patients

Three young Korean male patients aged 17, 31, and 34, with atrophic acne scarring on their cheeks, received treatment with PLA injection using a novel needle-free injector. All patients were in good health with no concurrent medical conditions and provided consent before the procedure. Before receiving the PLA injections, the patients had not previously undergone any treatments for their acne scars. In the pre-treatment consultation, we discussed the rationale for choosing PLA with a needle-free injector over conventional hyaluronic acid filler injections. The patients were informed that after PLA injection, dermal volume would increase gradually over several weeks due to the stimulation of collagen synthesis by dermal fibroblasts. The patients were also advised that they might experience temporary injection-related adverse effects such as swelling, redness, and petechiae, which were expected to resolve spontaneously.

### 4.2. Needle-Free Injection System

The needle-free injecting system (CUREJET; Baz Biomedic, Seoul, Republic of Korea) operates based on the following principle: an electric current applied to the actuator coil generates an electromagnetic force that propels the piston forward. When the piston contacts the membrane, the momentum is transferred to the liquid in the drug chamber, dispersing it from the nozzle onto the target area (Figure 2). Following the Venturi effect, the fluid propelled through the nozzle attains a very high speed, creating a microjet. Using Curejet, this process can be repeated at rates of up to 20 Hz.

### 4.3. Cadaver Study

To establish the treatment parameters, our initial step involved a preliminary cadaver study employing the same PLA product and injector (Figure 3).

Following a single injection, histological analysis indicated a uniform distribution of PLA particles across the upper- and mid-dermal layers, predominantly within the inter-collagen space, with a minor presence in the lower dermis and fat (Figure 4). There were no noticeable disruptions in the epidermal or vascular structures.

### 4.4. Treatment

Each patient underwent a single treatment. As a fibroblast-stimulating agent, a commercial PLA lyophilized powder (JUVELOOK; Vaim, Seoul, Republic of Korea) was employed, where one vial containing 42.5 mg of PDLLA and 7.5 mg of non-cross-linked HA was diluted with 8 mL of physiologic saline and vortexed for 2 h before administration. The treatment areas on the cheeks were cleansed with chlorhexidine and then topical anesthetic cream was applied for 20 min under occlusion. The reconstituted PLA suspension was administered into the base of atrophic scars using the CUREJET system. The treatment parameters were established following the outcomes of the cadaver study. The injector was set up with a 150 μm nozzle diameter, a frequency of 3 Hz, and an energy level of 5 (equivalent to 200 volts). Placing the nozzle at a 5 mm gap over the depressed scar area, the PLA suspension was injected into the dermis and subcutaneous fat through microjet injections. A minute volume of 1.5 microliters was administered for each shot. Two or three consecutive injections were administered to each atrophic scar lesion, delivering 1.5–2 mL of the product to each cheek. This approach was determined based on our preliminary study, which demonstrated a direct correlation between the quantity of drug delivered and the number of injections. The treatment endpoint was identifiable through the appearance of papules accompanied by focal blanching and central pinpoint bleeding, along with mild swelling observed at each injection site (Figure 5). The patients were evaluated 8–12 weeks after a single procedure and were further monitored up to six months.

## 5. Results and Discussion of Cases

All patients reported experiencing mild pain during the procedure, with pain scores ranging from 1 to 3 on a scale of 0 to 5. The patients encountered slight swelling and redness, lasting 24 to 36 h following the procedure. Mild petechiae developed in all patients but resolved spontaneously within 48 h. Two patients (Patients 1 and 3) presented with fine crusts at specific injection sites. These crusts spontaneously resolved within 3–4 days without leaving marks or post-inflammatory hyperpigmentation. No product-related adverse events, such as the formation of nodules or the development of local inflammation, were observed during the 6-month follow-up. A significant improvement in atrophic scarring was noted in all patients two to three months after a single treatment session (Figure 6).

Of note, atrophic scars in their early stages, accompanied by a slight redness, demonstrated rapid improvement in both height and volume after the single treatment (Figure 7).

The patients were highly satisfied with the improvements in their scars and also the positive changes in the texture and tone of their cheeks (Figure 8).

Our treatment focused on improving acne scars, which typically require deeper penetration. For this purpose, we selected the electromagnetic force-based needle-free injector over laser-assisted alternatives to effectively inject biostimulatory polymers at the base of acne scars. Our preclinical cadaver study revealed that the PLA particles were uniformly distributed throughout the full thickness of the dermis, as depicted in Figure 3. In a separate cadaver study conducted by Lee et al. [27], the researchers administered the same injectable using a laser-assisted needle-free injector, resulting in the distribution of the PLA confined primarily to the uppermost part of the dermal layer, indicating a more superficial penetration depth compared to our findings. This discrepancy confirms the impact of pressure generation methods on the depth of treatment when administering liquid jets through needle-free injections [34]. The heightened propulsion of liquid using electromagnetic force over laser-driven microbubbles might transiently release fibrosclerotic scar tissue, potentially allowing for the deeper distribution of PLA particles in our cases. While the primary mechanism of PLA injection in treating atrophic scars relies on the substance’s inherent fibroblast-stimulating properties, the impact of shock waves during microjet injection should also be considered. Given that improving atrophic acne scars requires the physical release of fibrosclerotic collagen bundles, the “acoustic subcision” effect [40] by shock waves could alleviate the mechanical burden of acne scar lesions. This potential contribution was observed in a recent randomized, controlled study focusing on the treatment of hypertrophic scars [41]. On the other hand, as shock waves are known to remodel the scar extracellular matrix in an anti-fibrotic manner [42], it is plausible that they might counteract the collagen synthesis induced by PLA, yet conclusive evidence validating this inference is still absent. Further research is suggested to elucidate the biomechanical impact of shock waves on treating atrophic scars.

In our patients, the improvement was not confined to the scars alone; there was an enhancement in the overall appearance of the skin. There has been limited clinical investigation in living human subjects to evaluate the skin rejuvenation effects following the application of PLA. Seo et al. [19] recently observed statistically significant differences in various signs of aging skin before and after the intradermal injection of PLA. Histologic examination revealed increased collagen and elastic fibers in the dermis. In a 24-week prospective, randomized, split-face study, the intradermal injection of PLA demonstrated increased the hydration and elasticity of the facial skin, along with improvements in facial skin texture, wrinkles, and pores [31]. These preliminary findings, along with the results in our patients, imply that injecting PLA into the dermis may contribute to overall skin quality improvement. The simultaneous application of intense pulsed light [43] or microneedle fractional radiofrequency [44] with PLA has demonstrated synergistic effects.

The outcomes of prior studies and clinical observations in our patients indicate that the needle-free microjet injector serves as a safe and advantageous tool for “collagen-boosting” procedures, effectively improving acne scarring and promoting skin enhancement and rejuvenation. Our protocol not only extends the advantages of the intradermal delivery of PLA using needle-free jet injectors to general skin rejuvenation but also broadens its benefits in improving atrophic scarring. The combined approach also offers reduced risks of side effects like visible nodule formation, aligning with recommendations from other researchers [31]. Moreover, the application of precisely controlled jet injectors can mitigate the risk of PLA-induced arterial obstruction by minimizing particle retrograde movement toward the major facial arteries, a serious complication associated with the heightened pressure of manual injection, as rarely reported in the literature [45].

One of the limitations of our study arises from our inability to accurately determine the volume of PLA injected. In our trial, the manufacturer of the device claims that it propelled approximately 1.5 microliters of liquid per shot, but we lack precise knowledge of the actual volume of PLA delivered into the dermis and subcutis. A similar needle-free jet injector achieved a delivery rate surpassing 90% of the fluid into the skin at depths of 3 mm in animals [33]. Some authors suggest that the presence of instant skin papules and remaining surface fluid corresponds to dermal drug deposition and serves as relevant clinical endpoints for needle-free jet injector treatments [46]. However, depending solely on subjective visual inspection renders it impossible to precisely assess the amount of drug delivered into the skin. Although the size of the bleb might indicate the injected volume, both its shape and the actual delivered volume can differ notably based on the viscosity of the injected drug [47]. For example, higher viscosity fluids do not spread laterally to the same extent as lower viscosity ones [48]. Finally, it is essential to acknowledge the primary limitation inherent in the cadaver skin model. In our cadaver study, the needle-free injection of PLA displayed an even distribution across the dermis. However, the consistency observed in cadaver skin in our preliminary study may not be replicated exactly in living human skin due to potential differences in its physical properties, particularly the viscoelasticity of the dermis. While our clinical cases demonstrated efficacy, the claimed advantage of accuracy with the needle-free injector remains unvalidated and warrants further experimental investigations for substantiation.

## 6. Conclusions

In conclusion, both the comprehensive literature review and our clinical cases highlight the application of an innovative needle-free injector in effectively addressing atrophic acne scars using PLA as a collagen-synthesis-boosting agent. The utilization of injectable PLA, especially when administered through advanced needle-free jet injectors, emerges as a secure and efficient alternative for rectifying atrophic acne scars. This approach may ensure consistent outcomes, irrespective of the clinician’s expertise; it also streamlines the procedure, diminishes patient discomfort, and reduces recovery time. These advantages collectively contribute to a more stable enhancement of scars and a rejuvenation effect of the PLA injection while minimizing associated side effects. Further research with a larger sample size and a controlled study design is recommended to confirm the safety and effectiveness of this treatment.

## Figures and Tables

**Figure 1 jcm-13-00440-f001:**
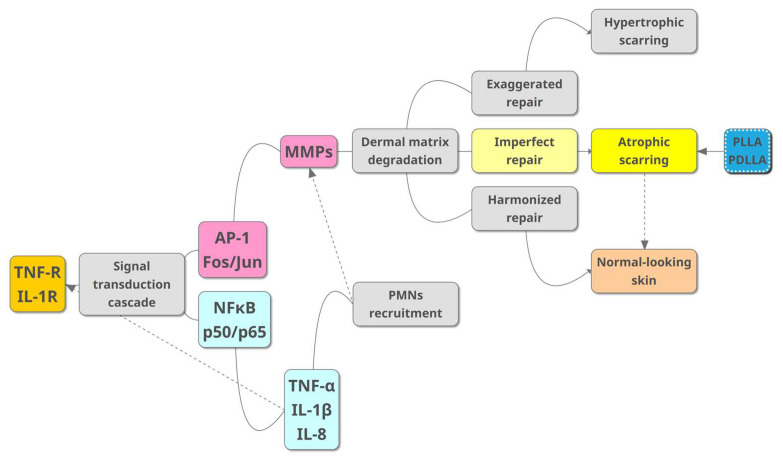
An illustration showing the current understanding of the pathophysiology of atrophic acne scarring and the role of poly-(lactic acid) to improve atrophic scars. Abbreviations: AP-1, activator protein 1; IL-1β, interleukin 1 beta; IL-1R, interleukin 1 receptor; IL-8, interleukin 8; MMP, matrix metalloproteinase; NF-κB, nuclear factor kappa-light-chain-enhancer of activated B cells; PDLLA, poly-(D,L-lactic acid); PLLA, poly-(L-lactic acid); PMN, polymorphonuclear neutrophil; TNF-α, tumor necrosis factor alpha; TNF-R, tumor necrosis factor receptor.

**Figure 2 jcm-13-00440-f002:**
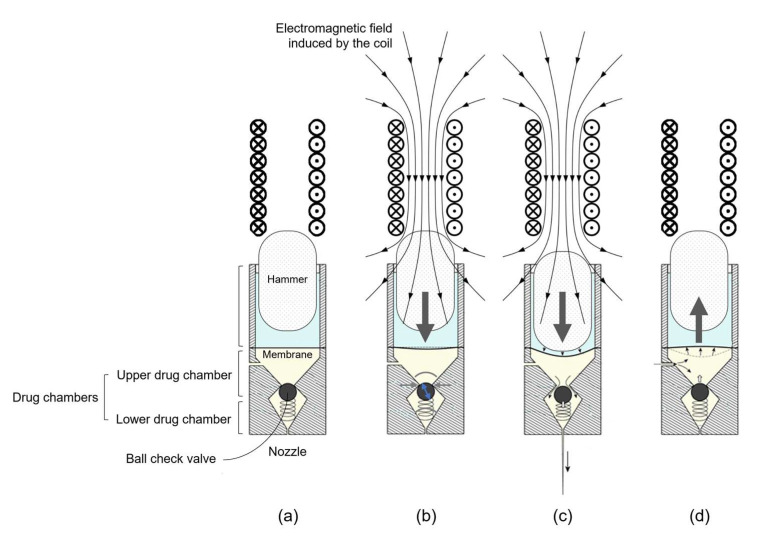
Schematic illustrations depicting the operational principle of the novel electromagnetic needle-free injector system utilized in this study. (**a**) Initial state; (**b**) The induction of electromagnetic fields in the coil causes a rapid linear movement of the hammer (piston), striking the inter-chamber membrane; (**c**) opening of the nozzle for injection of the liquid drug as a microjet, and (**d**) closure of the nozzle post injection, allowing the refilling of the chamber from the syringe.

**Figure 3 jcm-13-00440-f003:**
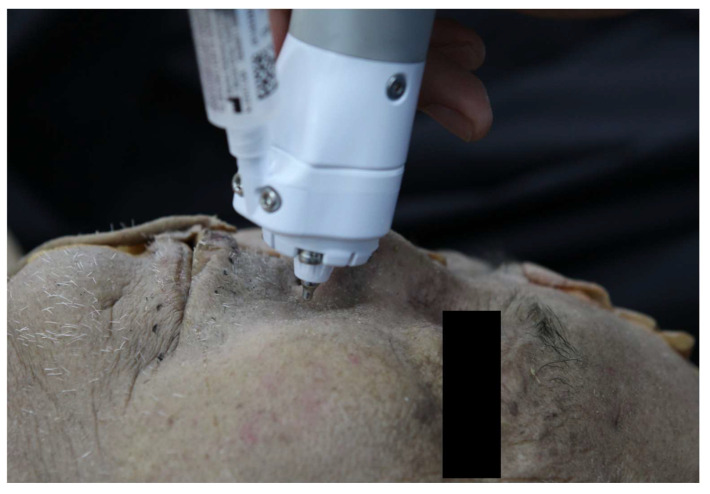
Demonstration of pre-clinical testing in an unfixed cadaver. The nozzle of the needle-free injector is positioned 5 mm from the facial skin surface.

**Figure 4 jcm-13-00440-f004:**
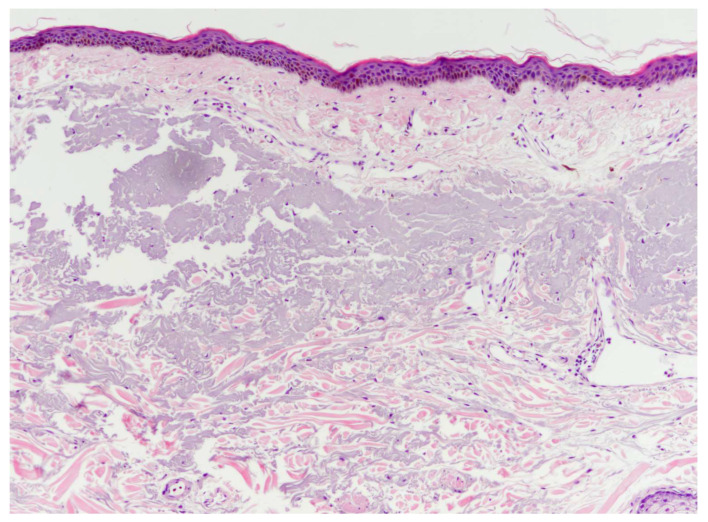
A histological examination of cadaver facial skin following a single injection using a novel needle-free injector, revealing a uniform distribution of poly-(lactic acid) particles in the dermis. These particles were primarily located in the upper- and mid-dermis, with a presence also in the lower dermis. The epidermal layer and blood vessels appear to be undisturbed. Hematoxylin and eosin stain (×100 magnification).

**Figure 5 jcm-13-00440-f005:**
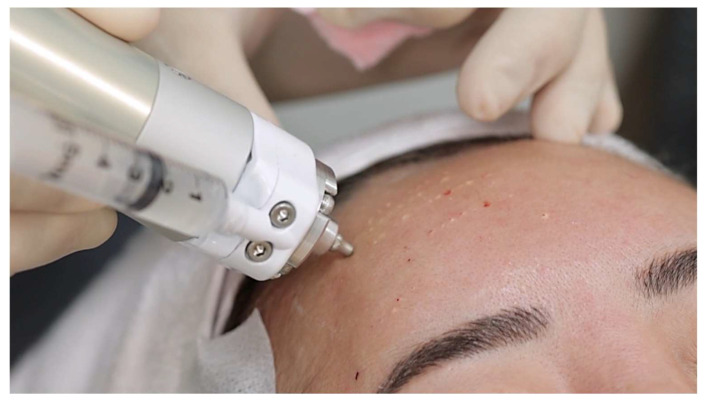
The treatment endpoint was marked by papules displaying focal blanching and central pinpoint bleeding, accompanied by mild swelling at each injection site. Image courtesy of Dr. Seong Jae Youn, Leaders Aesthetic Laser & Cosmetic Surgery Center, Seoul, Republic of Korea.

**Figure 6 jcm-13-00440-f006:**
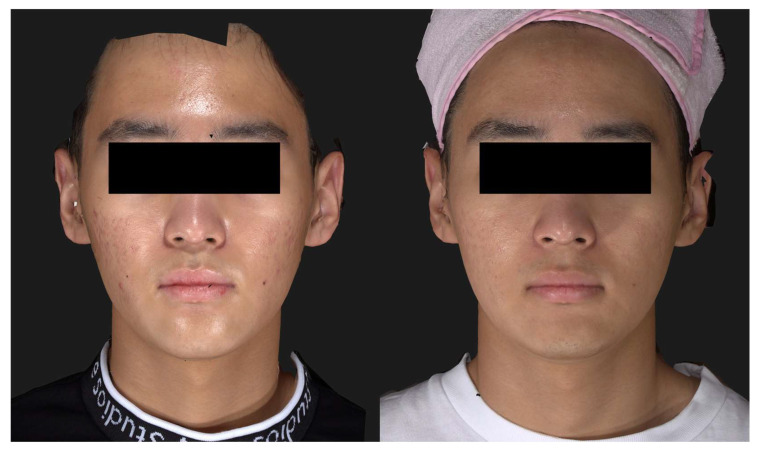
A notable improvement in atrophic scarring observed in Patient 1 (male, 17 years) after the injection of poly-(lactic acid) using a needle-free injector. Before (**left**) and two months after the single treatment session (**right**). Standardized clinical photos were taken with Vectra M3 (Canfield Scientific, Parsippany, NJ, USA).

**Figure 7 jcm-13-00440-f007:**
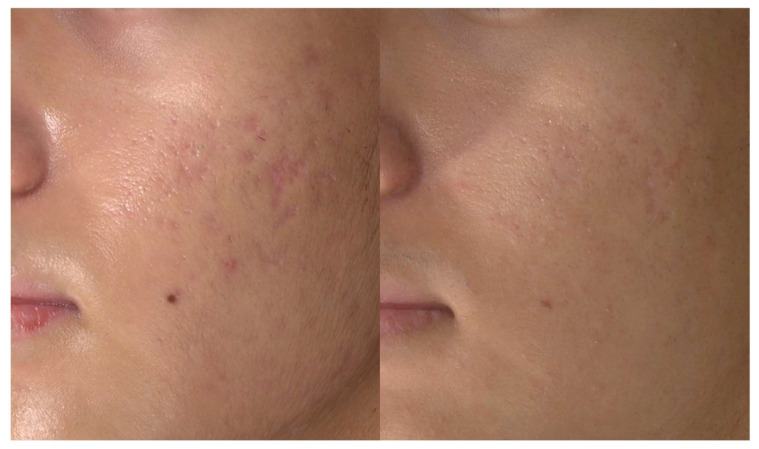
Close-up view of the acne scar lesions in the left cheek of Patient 1. Early depressed scars with slight erythema exhibited a rapid improvement in scar height and volume. Standardized clinical photos were taken with Vectra M3 (Canfield Scientific, Parsippany, NJ, USA).

**Figure 8 jcm-13-00440-f008:**
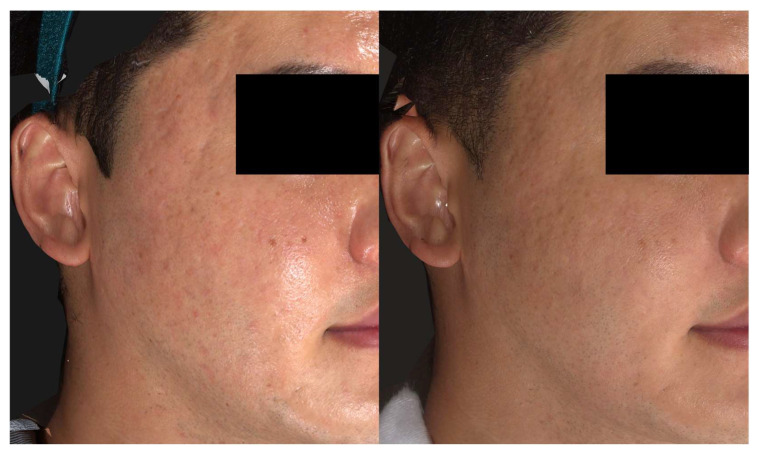
Enhanced texture and tone of facial skin, along with marked improvement in rolling acne scars in Patient 3 (male, 34 years), three months after a single treatment session. Standardized clinical photos were taken with Vectra M3 (Canfield Scientific, Parsippany, NJ, USA).

## Data Availability

Not applicable.

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
