# Peer review of "Needle-Free Jet Injection of Poly-(Lactic Acid) for Atrophic Acne Scars: Literature Review and Report of Clinical Cases"

_jcm, 2024, doi:10.3390/jcm13020440_

Round 1

Reviewer 1 Report

Comments and Suggestions for Authors

This study presents a new type of electromagnetic needle free syringe for intradermal injection of polylactic acid (PLA) for the treatment of acne atrophic scars. Has certain research significance, and puts forward several suggestions:

1. Needle free injection systems have been studied in the field of dermatology, and generally, skin papules and surface fluid residues may occur. How to evaluate how much of the injected drug enters the dermis of the skin, with only slight swelling and ecchymosis described in the text?

2. How to determine the clinical treatment endpoint? Providing a picture of the treatment endpoint as a basis would be more convincing.

3. In the discussion, the author proposed that electromagnetic driven needle less injection technology is superior to laser driven technology, but there is no data in this manuscript to support this assertion.

4. As the authors have stated, the occurrence of skin nodule reactions after injection is a thorny complication of intradermal injection of PLLA. How can a needle free syringe avoid such reactions?

Author Response

A point-by-point response to the reviewer’s comments (reviewer 1, round 1)

We would like to thank the reviewers for their thoughtful comments and efforts towards improving our manuscript. In the following, we present our response specific to each reviewer comments.

[Reviewer’s comment] 1) Needle free injection systems have been studied in the field of dermatology, and generally, skin papules and surface fluid residues may occur. How to evaluate how much of the injected drug enters the dermis of the skin, with only slight swelling and ecchymosis described in the text?

[Authors’ response] Thank you very much for your feedback. As the reviewer clearly mentioned, our original submission is not describing the estimation of the injected drug into the dermis. We admit that this is really the main weakness of the present manuscript. We acknowledge the reviewer's criticism regarding the limitations associated with estimating the delivered drug volume solely through subjective visual inspection. While the size of the injection bleb might suggest the injected volume, its shape varies significantly based on the viscosity of the drug, as Simmons et al. (2019) highlight that higher viscosity fluids don't spread laterally to the same extent as lower viscosity ones. In our trial, the jet injector propelled 1.5 microliters of liquid per shot, yet we lack precise knowledge regarding the actual volume of PLA delivered into the dermis and subcutis. This limitation is openly added at the end of the Discussion section, with corresponding references, as: “One of the limitations of our study arises from our inability to accurately determine the volume of PLA injected. In our trial, the manufacturer of the device claims it propelled approximately 1.5 microliters of liquid per shot, but we lack precise knowledge of the actual volume of PLA delivered into the dermis and subcutis A similar needle-free jet injector achieved a delivery rate surpassing 90% of the fluid into the skin at depths of 3 mm in animals. Some authors suggest that the presence of instant skin papules and remaining surface fluid corresponds to dermal drug deposition and serves as relevant clinical endpoints for needle-free jet injector treatments. However, depending solely on subjective visual inspection renders it not possible to precisely assess the amount of drug delivered into the skin. Although the size of the bleb might indicate the injected volume, both its shape and the actual delivered volume can differ notably based on the viscosity of the injected drug. For example, higher viscosity fluids do not spread laterally to the same extent as lower viscosity ones”.

[Reviewer’s comment] 2) How to determine the clinical treatment endpoint? Providing a picture of the treatment endpoint as a basis would be more convincing.

[Authors’ response] We appreciate the reviewer’s valuable suggestion. We added a new figure (Figure 4) which describes the immediate post-injection skin endpoint. Also, we put a new sentence describing the endpoint in the “Treatment” section of the manuscript: “The treatment endpoint was identifiable through the appearance of papules accompanied by focal blanching and central pinpoint bleeding, along with mild swelling observed at each injection site”. Figures in the original submission were re-numbered, accordingly.

[Reviewer’s comment] 3) In the discussion, the author proposed that electromagnetic driven needle less injection technology is superior to laser driven technology, but there is no data in this manuscript to support this assertion.

[Authors’ response] Thank you very much to point out the weakness of our manuscript. We wish to disclose that we recognize both types of jet injectors possess advantages and weaknesses. Our manuscript's statement does not aim to assert the superiority of electromagnetic jet injectors over laser-based systems. To the best of the authors' knowledge, there are presently no investigational or clinical studies directly comparing the efficacy and safety of two types of needle-free jet injectors: electromagnetic versus laser-driven systems. However, concerning injection depth and delivery volume, the electromagnetic system may be deemed more suitable, while for precise injection control, preference may lean towards laser-driven jet injectors. In terms of cost-effectiveness, laser-based jet injectors have a drawback due to the considerably higher expense of the laser system compared to electromechanical actuators. We added some new sentences (with corresponding references) in Discussion section: “However, both electromagnetic-based and laser-driven jet injectors possess advantages and disadvantages in comparison to each other. In relation to injection depth and delivery volume, the electromagnetic system may be considered more appropriate, whereas for precise injection control, the preference might shift towards laser-driven jet injectors. Regarding cost-effectiveness, laser-based jet injectors are disadvantaged due to the significantly higher cost of the laser system compared to electromechanical actuators”.

[Reviewer’s comment] 4) As the authors have stated, the occurrence of skin nodule reactions after injection is a thorny complication of intradermal injection of PLLA. How can a needle free syringe avoid such reactions?

[Authors’ response] We are grateful for the reviewer's significant inquiry and have incorporated additional sentences, supported by referenced material, to further enhance the content.: “It is widely acknowledged that precision in PLA injections into facial skin is crucial as even minor errors like uneven material distribution can lead to irregular outcomes. Researchers suggest using the lowest volumes as possible (e.g., injections of less than 0.05 mL) of more diluted injectable PLA to avoid such problems. Given that needle-free jet injectors are engineered to deliver precise drug amounts (e.g., 0.0015 ml per shot in the authors’ device) to a specific depth, the anticipated risk of implant nodule formation with this system might be considerably lower compared to manual syringe injections”. Accordingly, the old sentence was removed from the manuscript: “In this context, using needle-free microjet injectors is also advantageous in reducing the risk of nodule formation compared to manual injection, as they enable precise filler delivery to a specific depth”.

In response to the request from [reviewer 2] to transform the manuscript into a literature review on PLA injection techniques, incorporating personal case experiences, substantial revisions and expansion were implemented. The revised manuscript now follows a "literature review and case presentation" structure, outlining the contemporary insights into the utilization of PLA injection for acne scar treatment and the advancements in needle-free injection systems. The second segment of the manuscript details the clinical cases involving the treatment of atrophic acne scars, employing a combined approach integrating PLA and a needle-free injector. As a result of these extensive changes, the Abstract was revised to align with the modifications. Additionally, the manuscript title was changed according to reviewer 2's suggestions. The revised title is as follows: “Needle-Free Jet Injection of Poly-(Lactic Acid) for Atrophic Acne Scars: Literature Review and Report of Clinical Cases”. Also, we have incorporated two additional figures: one providing a comprehensive overview of the pathophysiology of acne scarring, and the other visualizing the clinical endpoint of the procedure. We kindly seek understanding from reviewer 1 for many modifications made.

Again, we would like to show our deepest appreciation regarding the reviewer’s valuable comments and suggestions.

Nark-Kyoung Rho (the corresponding author)

Reviewer 2 Report

Comments and Suggestions for Authors

This is a presentation of the new injecting device for PLA. 

In my opinion the paper might be interesting but have some flaws. This study would  be proper if the study group would be extended. What PLA you used in this presentation ?

please clarify more what is a Novelty because this is missing in the paper. 

How can you derived clear conclusions if you are not able to present pain scale results like VAS before and after the treatment however this results should be compared to the results of other participants with other injections. 

recerence section is too short - i suggest to rebuild the paper completely e.g. As a review of the literature about PLA injections techniques as a Strong fundament and after that of before that you should make short presentation of this case/technique. Title should be changed. 

Paper is too short reference section is not good. 

Rebuild the manuscript. Major revisions are needed. 

Best Regards for the Authors and I hope that I can see a new version of this manuscript after major revisions 

Author Response

A point-by-point response to the reviewer’s comments (reviewer 2, round 1)

We would like to thank the reviewers for their thoughtful comments and efforts towards improving our manuscript. In the following, we present our response specific to each reviewer comments.

[Reviewer’s comment] 1) In my opinion the paper might be interesting but have some flaws. This study would be proper if the study group would be extended.

[Authors’ response] Thank you very much for your feedback. We acknowledge the limitation of a small patient sample in our study, which prompted us to submit this manuscript as a "Case Report." We agree on the necessity for further studies involving a larger number of patients to substantiate the claims made in the current manuscript.

[Reviewer’s comment] 2) What PLA you used in this presentation?

[Authors’ response] As we disclosed in the Treatment section, a commercial PLA lyophilized powder (JUVELOOK; Vaim, Seoul, Korea) was employed, where one vial containing 42.5 mg of poly-(d,l-lactic acid) and 7.5 mg of non-cross-linked HA.

[Reviewer’s comment] 3) Please clarify more what is a Novelty because this is missing in the paper.

[Authors’ response] We appreciate the reviewer’s valuable comment. We admit that the original submission of this simple case series fails to highlight a novelty in the field of aesthetic medicine and surgery. Based on the limited evidence available, prior reports on PLA jet injection primarily focused on general skin rejuvenation, targeting wrinkles, pores, and texture improvement. Our cases' results supplement these findings by showcasing improvements in atrophic acne scars, expanding the aesthetic applications of needle-free intradermal delivery of PLA. Consequently, we have included sentences emphasizing this aspect, along with additional benefits of the presented protocol: “Our protocol not only extends the advantages of intradermal delivery of PLA using needle-free jet injectors to general skin rejuvenation but also broadens its benefits in improving atrophic scarring. The combined approach also offers reduced risks of side effects like visible nodule formation, aligning with recommendations from other researchers. Moreover, the application of precisely controlled jet injectors can mitigate the risk of PLA-induced arterial obstruction by minimizing particle retrograde movement toward the major facial arteries, a serious complication associated with the heightened pressure of manual injection, as rarely reported in the literature”.

[Reviewer’s comment] 4) How can you derived clear conclusions if you are not able to present pain scale results like VAS before and after the treatment however this results should be compared to the results of other participants with other injections.

[Authors’ response] We apologize for the oversight in our conclusion, as we did not include a comparative analysis of injection pain among different injection techniques in this case series. Given the nature of our study, which is a case series, we were unable to directly compare pain levels between jet injectors and manual injections. Nonetheless, we incorporated findings from previous studies conducted by other researchers that addressed the comparison of pain levels. We added a new paragraph in the Discussion section: “One of the most significant advantages of needle-free pressure injectors compared to manual injections could be their comparatively lower pain levels. In a recent comparative study, 17 patients received injections with a needle, while 22 patients underwent jet injector injections. The study showed significant differences in patient-reported pain levels during the procedure (p<0.005), with those in the jet injector group experiencing less pain than those receiving needle injections”.

[Reviewer’s comment] 5) I suggest to rebuild the paper completely e.g. As a review of the literature about PLA injections techniques as a Strong fundament and after that of before that you should make short presentation of this case/technique. / Paper is too short reference section is not good. / Rebuild the manuscript. Major revisions are needed.

[Authors’ response] Your valuable recommendations are greatly appreciated. The authors have thoroughly considered the suggestion to convert the manuscript into a narrative review of the literature, as proposed by the reviewer. The revised manuscript now follows a "literature review and case presentation" structure, outlining the contemporary insights into the utilization of PLA injection for acne scar treatment and the advancements in needle-free injection systems. The second segment of the manuscript details the clinical cases involving the treatment of atrophic acne scars, employing a combined approach integrating PLA and a needle-free injector. Also we have incorporated an additional figure providing a comprehensive overview of the pathophysiology of acne scarring.

[Reviewer’s comment] 6) Title should be changed.

[Authors’ response] Thank you very much for the suggestion. The authors discussed on changing the title of the manuscript. The revised title is as follows: “Needle-Free Jet Injection of Poly-(Lactic Acid) for Atrophic Acne Scars: Literature Review and Report of Clinical Cases”

Due to many changes made in the manuscript, we have revised the Abstract to conform to the revision.

Again, we would like to show our deepest appreciation regarding the reviewer’s valuable comments and suggestions.

Nark-Kyoung Rho (the corresponding author)

Round 2

Reviewer 1 Report

Comments and Suggestions for Authors

The authors have made revisions as requested by the reviewers. Authors point out in the abstract that injection PLLA are evenly distributed in all layers of the dermis in cadaver skin, and this histological evidence supports the clinical efficacy of needle free injection, believing that needle free injection has advantages in accuracy and comfort. However, the elasticity of cadaver skin is completely different from that of living skin, and this uniform distribution may not necessarily be true in living organisms. Although clinical efficacy has been shown to be effective, the accuracy advantage of needleless syringes is questionable and requires more preliminary experimental support.

Author Response

A point-by-point response to the reviewer’s comments (reviewer 1, round 2)

We would like to thank the reviewers for their thoughtful comments and efforts towards improving our manuscript. In the following, we present our response specific to the comments from the second-round review.

[Reviewer’s comment] 1) Authors point out in the abstract that injection PLLA are evenly distributed in all layers of the dermis in cadaver skin, and this histological evidence supports the clinical efficacy of needle free injection, believing that needle free injection has advantages in accuracy and comfort. However, the elasticity of cadaver skin is completely different from that of living skin, and this uniform distribution may not necessarily be true in living organisms. Although clinical efficacy has been shown to be effective, the accuracy advantage of needleless syringes is questionable and requires more preliminary experimental support.

[Authors’ response] We appreciate the insightful feedback once again. The reviewer rightly highlighted the potential differences in physical properties between cadaver skin and living human skin, though direct comparative studies seem to be scarce. This limitation is openly added at the end of the Discussion section, as: “Finally, it is essential to acknowledge the primary limitation inherent in the cadaver skin model. In our cadaver study, the needle-free injection of PLA displayed an even distribution across the dermis. However, the consistency observed in cadaver skin in our preliminary study may not replicate exactly in living human skin due to potential differences in its physical properties, particularly the viscoelasticity of the dermis. While our clinical cases demonstrated efficacy, the claimed advantage of accuracy with the needle-free injector remains unvalidated and warrants further experimental investigations for substantiation”.

Again, we would like to show our deepest appreciation regarding the reviewer’s valuable comments and suggestions.

Nark-Kyoung Rho (the corresponding author)

Reviewer 2 Report

Comments and Suggestions for Authors

Accept as it is 

Author Response

A point-by-point response to the reviewer’s comments (reviewer 2, round 2)

We would like to thank the reviewers for their thoughtful comments and efforts towards improving our manuscript. In the following, we present our response specific to the comments from the second-round review.

[Reviewer’s comment] 1) Accept as it is.

[Authors’ response] We are grateful for your feedback. The valuable suggestions offered by the reviewers in the first round of review have immensely contributed to refining our manuscript.

In response to the advice of the reviewer 1 (second round), we have incorporated several sentences to conclude the Discussion section., as: “Finally, it is essential to acknowledge the primary limitation inherent in the cadaver skin model. In our cadaver study, the needle-free injection of PLA displayed an even distribution across the dermis. However, the consistency observed in cadaver skin in our preliminary study may not replicate exactly in living human skin due to potential differences in its physical properties, particularly the viscoelasticity of the dermis. While our clinical cases demonstrated efficacy, the claimed advantage of accuracy with the needle-free injector remains unvalidated and warrants further experimental investigations for substantiation”.

Again, we would like to show our deepest appreciation regarding the reviewer’s valuable comments and suggestions.

Nark-Kyoung Rho (the corresponding author)